# Evaluation of Two Different Preparation Protocols for MALDI-TOF MS Nontuberculous Mycobacteria Identification from Liquid and Solid Media

**DOI:** 10.3390/microorganisms11010120

**Published:** 2023-01-03

**Authors:** Lisa Pastrone, Antonio Curtoni, Giulia Criscione, Francesca Scaiola, Paolo Bottino, Luisa Guarrasi, Marco Iannaccone, Markus Timke, Cristina Costa, Rossana Cavallo

**Affiliations:** 1Microbiology and Virology Unit, University Hospital Città della Salute e della Scienza di Torino, 10126 Turin, Italy; 2Department of Public Health and Paediatrics, University of Turin, 10126 Turin, Italy; 3Bruker Daltonics GmbH & Co. KG, 28359 Bremen, Germany

**Keywords:** NTM, mycobacteria, MALDI-TOF MS, identification, LPA, 7H11, MGIT

## Abstract

Nontuberculous mycobacteria (NTM) identification is essential for establishing the relevance of the isolate and for appropriate antimicrobial therapy. Traditionally, NTM identification is performed by using Line Probe Assays (LPA), a costly and time-consuming technique requiring trained personnel. MALDI-TOF MS is a promising tool for NTM identification, and its use is rapidly growing. We evaluated the newly introduced MBT Mycobacteria kit (MBT) and the MycoEx preparation protocol (Bruker Daltonics, Germany) for NTM MALDI-TOF MS identification using LPA results as a reference. Fifty NTM grown on 7H11 agar and MGIT broth were analyzed with both protocols using the Bruker Microflex^®^ LT MALDI-TOF MS (Bruker Daltonics) instrument. MBT and MycoEx provided identification results in 97.0% and 95.0% of the cases, respectively. With both protocols, 100% of the provided results agreed with LPA with no registered mismatch. MBT achieved an elevated number of highly probable identifications (88.0% vs. 83.0%) and a higher reproducibility rate of correct results (86.6% vs. 75.8%) in comparison to MycoEx. This study provides results about MBT performance for liquid and solid media, underlining the strengths and weakness under different conditions. Our results suggest that MALDI-TOF MS could provide a great advantage for timely and cost-saving NTM identification with potential implications for patient outcome.

## 1. Introduction

The genus *Mycobacterium* includes a high number of species (more than 190) divided into the following three groups: *Mycobacterium tuberculosis* complex, *Mycobacterium leprae* and Nontuberculous mycobacteria (NTM), also named atypical or environmental mycobacteria [1]. NTM are opportunistic pathogens for humans, and are widespread in the environment, such as soil and water. They are aetiologic agents of mycobacteriosis and predisposing factors, such as comorbidities, coinfections, immunosuppression and advanced age, are involved in the disease onset [2]. They occur as lung, skin, soft tissue, lymphatic and disseminated infections, as well as outbreaks in hospital settings related to inadequate medical device disinfection or sterilization [3]. Mycobacteriosis incidence and impact have risen worldwide, and NTM show a high morbidity and mortality rate due to diagnostics and therapeutic issues [4].

An early and accurate diagnosis of mycobacterial infections is pivotal in the clinical management of patients and for treatment options. An inappropriate antimicrobial therapy can negatively impact patient outcome due to exposure to toxic drugs and the selection of multi-resistant strains [5]. Thus, NTM identification at different levels (group, species and subspecies) is important in order to define the clinical relevance of isolates and to establish the appropriate therapy when needed. Even the most relevant international societies of Respiratory Diseases, Infectious Diseases and Clinical Microbiology (ATS/ERS/ESCMID/IDSA) underline the usefulness of NTM identification in clinical settings [6].

The gold standard test for NTM identification is gene sequencing. Unfortunately, the access to this technique is currently confined to a limited number of laboratories due to its costs, the scant availability of qualified staff with dedicated professional roles and the long response times [7,8].

Clinical Microbiology Laboratories (CML) routinely perform NTM identification with Line Probe Assays (LPA), a genotyping method based on DNA sequences detection through hybridization with nucleotide probes fixed on nitrocellulose strips. However, even this technique is characterized by long turn-around times (TAT), high costs and the need for specifically trained personnel [9].

Among new technologies, Matrix Assisted Laser Desorption Ionization-Time of Flight Mass Spectrometry (MALDI-TOF MS), currently used in CML for bacteria and fungi identification, could represent a powerful tool for NTM detection. This proteomics-based technology facilitates the rapid recognition of microorganisms based on unique spectral fingerprints produced by extracted proteins derived from colonies grown in solid and liquid culture media [10]. Given the excellent performance compared with conventional methods, MALDI-TOF MS use in Mycobacteriology is rapidly growing [11]. However, unlike most microorganisms, mycobacteria require a dedicated preparation treatment. Cell inactivation and protein extraction phases are essential before analysis by MALDI-TOF MS, due to the biosecurity issues and complex structural features of mycobacterial cell wall [12]. To resolve this issue, several protocols based on chemical and mechanical methods were proposed [13,14]. Among these techniques, the MycoEx preparation protocol (MycoEx) (Bruker Daltonics, Bremen, Germany) is a widespread protocol for Mycobacteria spectra acquisition on Bruker MALDI-TOF platforms and it is based on thermal cell inactivation. Unfortunately, the MycoEx protocol requires several manual steps and the heat-inactivation procedure could expose laboratory personnel to greater risks associated with contaminated aerosol.

The new ready-to-use kit MBT Mycobacteria kit (MBT) (Bruker Daltonics, Germany), based on chemical cell-inactivation, could overcome the MycoEx protocol’s disadvantages for rapid mycobacteria identification by MALDI-TOF-MS. The study aim was to compare MycoEx and MBT for MALDI-TOF MS identification of the principal NTM strains of clinical interest. Both techniques were tested from positive liquid broth and a solid medium in order to define the optimized identification solution. LPA results were used as reference.

## 2. Materials and Methods

The study was performed in the Microbiology and Virology Unit (University Hospital “A.O.U. Città della Salute e della Scienza di Torino”, Turin—Italy) between October 2021 and March 2022. Fifty NTM clinical isolates, previously collected from different types of specimens (mainly respiratory) and belonging to 6 different species were selected from laboratory strains stored at −80 °C. In particular, 3 rapid growing mycobacteria (RGM), *M. abscessus subsp. abscessus*, *M. abscessus subsp. massiliense*, and *M. fortuitum*, and 4 slow growing mycobacteria (SGM), *M. avium*, *M. chimaera*, *M. intracellulare*, and *M. gordonae* were considered.

In agreement with our laboratory Standard of Care (SoC), all NTM positive cultures were identified by molecular genotypic methods. The LPA commercial system, the most reliable tool for mycobacteria identification for labs not performing sequencing, was used. DNA extraction was carried out with the GenoLyse kit v. 1.0 (Hain Lifescience GmbH, Nehren, Germany). The amplification and hybridization for NTM species-level identification were performed with GenoType Mycobacterium CM v. 2.0 or GenoType Mycobacterium AS v. 1.0 and GenoType NTM-DR v. 2.0 (Hain Lifescience GmbH), if required, according to the manufacturer’s instructions. Amplification and hybridization steps were performed with the Veriti™ Thermal Cycler (Applied BioSystem, Waltham, MA, USA) and TwinCubator (Hain Lifescience GmbH) instruments, respectively. The estimated TAT could range from 4 to 6 h on the basis of the considered mycobacterium and of the group, species, and subspecies identification level.

The selected strains were thawed and cultured in BBL™ MGIT broth (MGIT) (Becton, Dickinson and Company, Franklin Lakes, NJ, USA) and incubated at 37 °C in BACTEC™ MGIT™ 960 (Becton, Dickinson and Company) until positive signal detection; at the same time, they were seeded on Middlebrook 7H11 agar (Becton, Dickinson and Company) and incubated in 5% CO_2_ atmosphere at 37 °C up to NTM evidence as visible colonies. Therefore, positive samples from MGIT and 7H11 agar were processed in parallel with the new MBT Mycobacteria kit and the MycoEx protocol according to the manufacturer’s instructions.

### 2.1. MBT Mycobacteria Kit Procedure

A mycobacterial aliquot was washed and inactivated with the provided reagents (Washing Solution, Inactivation Reagent) for 30 min at room temperature. Then, each sample was combined with acetonitrile (100 μL for 7H11 agar, 50 μL for MGIT), and 70% formic acid (100 μL for 7H11 agar, 50 μL for MGIT) and centrifuged at 13.000 rpm for 2 min. Finally, 1 µL of supernatant was spotted in triplicate onto the MALDI plate. The estimated TAT both on MGIT and 7H11 agar was about 40 min.

### 2.2. MycoEx Preparation Protocol

At the same time, the MycoEx protocol was performed as follows: after an initial heat-inactivation treatment (100 °C for 30 min), samples were processed with absolute ethanol (900 μL) and centrifuged at 13,000 rpm for 2 min. Then, the pellet was dried and resuspended with zirconia microsphere (40 mg), acetonitrile (25 μL) and 70% formic acid (25 μL) in order to break the mycobacterial cell wall. Finally, samples were centrifugated at 13,000 rpm for 2 min and, as mentioned above, 1 µL of supernatant was spotted in triplicate onto the MALDI-TOF MS plate. In this case, the estimated TAT both on MGIT and 7H11 agar was about 1 h.

### 2.3. MALDI-TOF MS Spectra Acquisition and Analysis

An extract of *Escherichia coli* proteins (Bacterial Test Standard, BTS, Bruker Daltonics) was used as internal quality control by spotting two replicates onto the MALDI-TOF MS plate.

For each sample, spectra were acquired with the Bruker Microflex^®^ LT MALDI-TOF MS (Bruker Daltonics) instrument and analyzed with the latest release of IVD MBT Mycobacteria Library v.4.0 (Bruker Daltonics). Spectra were acquired in a linear positive ion mode at a laser frequency of 60 Hz across a mass/charge ratio (m/z) of 2000 to 20,000 Da.

Interpretation of results was performed according to the manufacturer’s instructions. NTM identification was considered highly probable with a *log*(score) ≥ 1.80, as probable with *log*(score) between 1.60 and 1.79, and not assigned with *log*(score) < 1.60. Mycobacterial identification was considered correct at the species/complex level if at least one of the three replicates agreed with the SoC results with *log*(score) ≥ 1.60. Identifications of species closely related inside the *M. abscessus* complex, *M. fortuitum* group and *M. chimaera-intracellulare* group were considered concordant.

### 2.4. Statistical Analysis

For both extraction methods (MycoEx protocol and MBT Mycobacteria kit), we described *log*(score) distribution, and we compared the effects of the extraction methods on the frequencies of samples without identification using Fisher’s exact test. As an exploratory analysis, we tested the effects of the type of culture media and of the rapid and slow mycobacteria growth rate on the frequencies of samples without identification. Moreover, the reproducibility rate of the triplicate NTM identification results for both testing protocols was assessed.

## 3. Results

Both the MBT Mycobacteria kit and MycoEx protocol had a higher proportion of highly probable identification results, *log*(score) ≥ 1.80, with an overall rate of 88.0% (95% CI 80.0–93.2) and 83.0% (95% CI 74.4–89.2), respectively. Considering the growing media, the MBT kit and MycoEx protocol reached 94.0% (95% CI 83.2–98.6) and 84.0% (95% CI 71.2–91.9) in 7H11 agar, respectively, and both reached 82.0% in MGIT broth (see Table 1).

Positive identification results were achieved in 97.0% (95% CI 91.5–99.4) and 95.0% (95% CI 88.7–98.4) of the cases with the MBT kit and MycoEx protocol, respectively. Taking into account the type of culture media, the MBT kit provided identification results in 98.0% (95% CI 89.4–99.9), and in 96.0% (95% CI 86.3–99.5) on 7H11 agar and MGIT, respectively, while the MycoEx protocol provided identification results for 92.0% (95% CI 80.8–97.8) and 98.0% (95% CI 89.4–99.9) (see Figure 1).

The MBT kit and MycoEx protocol reached 100% agreement in the samples positive for *M. chimaera*, *M. intracellulare*, and *M. gordonae* regardless of the starting growth media. In addition, the MBT kit correctly identified all the samples positive for the *M. fortuitum* group (see Figure 2).

The MBT kit did not identify NTM in three (3%) cases and MycoEx protocol in five (5%). Two (2%) samples, 1 M. avium and 1 M. abscessus complex, were not identified by both extraction methods on the same media. The details of the unidentified NTM according to the extraction protocol and the culture media are reported in Table A1 in Appendix A.

We did not observed differences in frequencies of samples without results both overall (Fisher’s exact test *p*-value = 0.721) and according to the growing media. In 7H11 agar, the number of unidentified samples was five (one extracted with the MBT kit and four extracted with the MycoEx protocol; Fisher’s exact test *p*-value = 0.362) and in MGIT it was three (two extracted with the MBT kit and one extracted with the MycoEx protocol; Fisher’s exact test *p*-value = 1.000).

Among the identified NTM, no mismatches were observed for both protocols; therefore, no further in-depth analysis was performed.

Focusing on the reproducibility of the triplicate tests, the MBT kit resulted in a higher rate of 2 on 3 and 3 on 3 probable identifications (*log*(score) ≥ 1.60) when compared with the MycoEx protocol (Fisher’s exact test *p*-value = 0.066): 94.8% (95% CI 88.2–98.1) vs. 91.6% (95% CI 84.0–95.9), respectively. Concerning 3 on 3 probable identifications, the MBT kit’s analytical reproducibility rate was 86.6% (95% CI 78.3–92.1) *vs*. 75.8% (95% CI 66.2–83.4) for the MycoEx protocol (Fisher’s exact test *p*-value = 0.359).

Reproducibility results according to the preparation kit or protocol, *log*(score) and type of mycobacteria and growing media are summarized in Table A2 in Appendix A.

## 4. Discussion

NTM identification, usually based on commercial hybridization methods, is limited to less than 40 species. Moreover, LPA assays are expensive and time consuming and could require multiple additional sessions to distinguish between mycobacterial species, especially inside groups or subspecies [15]. The implementation of MALDI-TOF MS in CML markedly improved bacterial identification by increasing diagnostic accuracy, the speed of diagnosis and reducing time and costs [16]. For NTM, MALDI-TOF MS mycobacteria libraries overcome LPA in terms of potential detectable species, reaching a number of more than 180 different species in the MBT Mycobacteria library (Bruker Daltonics), one of the most wide-ranging MALDI-TOF MS Mycobacteria library. However, MALDI-TOF MS mycobacteria identification is still considered challenging and not commonly performed [17]. Its difficulties are related to mycobacteria cell wall composition and the propensity to form aggregates, which does not permit the use of conventional MALDI-TOF MS protein extraction methods [18]. To obtain high quality spectra and reliable mycobacteria identification, different MALDI-TOF MS preparation protocols based on heat, chemical and mechanical treatments have been developed and achieved good performance in terms of agreement with reference methods [13,14,19,20,21,22].

Our work focused on the evaluation of two different preparation methods, namely the recently introduced MBT Mycobacteria kit and the MycoEx protocol, for NTM spectral MALDI-TOF MS identification on Bruker MALDI-TOF MS platforms, from MGIT broth and 7H11 solid medium towards our SoC, LPA assays. To our knowledge, this is the first pilot study estimating the new MBT Mycobacteria kit’s MALDI-TOF MS identification performance and defining its potential advantages and disadvantages for application in the clinical microbiological routine workflow.

### 4.1. Protocol Identification Performance Considerations

The *log*(score) analysis highlights a better performance of the MBT kit in producing results with a high probability of correct identification. The difference in the achievements was independent from RGM or SGM groups but was linked to the growth on solid 7H11 medium.

The MBT kit, in comparison to the MycoEx protocol, also provided successful identifications in more cases with the major advantage in its application on positive 7H11 solid media. Moreover, the MBT kit achieved completely successful results in 4 of 6 of the clinically significant considered species and groups (*M. chimaera*, *M. intracellulare*, *M. gordonae*, *M. fortuitum* group) while the MycoEx protocol failed in achieving 100% success for the *M. fortuitum* group. Identifications were correct for all the examined clinically isolated strains of NTM with both methods and media, showing a good agreement with the LPA reference technique. Interestingly, no mismatches were observed for all tested samples, confirming MALDI-TOF MS as an effective and reliable method for NTM identification regardless of the adopted extraction protocol. Overall, these results are in line with those reported in the literature, even if a fair comparison could present bias due to the different instruments, libraries, identification criteria, number of replicates, extraction protocols, media and groups/species considered [15,16,23,24,25].

For both the MBT Mycobacteria kit and MycoEx extraction protocol, the replicate analysis of samples, according to other studies, was pivotal in order to obtain better results [16,17,19,20,21]. The importance of running replicate tests was also supported by a recent multicentre analysis, performed on 1330 mycobacterial strains collected from twelve CML [23]. NTM identification at the complex and species level was higher at the involved laboratories in the case of duplicates tests [23]. The MBT kit showed better performance when compared with the MycoEx protocol in terms of both duplicate and triplicate tests, with probable and highly probable results; therefore, samples with fewer replicates are required to obtain correct identifications.

### 4.2. Turn-Around Time and Cost-Saving Considerations

In terms of the TAT evaluation, both protocols registered a significantly shorter TAT compared with our laboratory SoC procedure (about 5 h). In addition, the MBT kit’s protocol process (five main steps) and TAT (about 40 min) were lower even than those for MycoEx (seven main steps and around 1 h), suggesting that its adoption could be less time-consuming and labor-intensive.

MALDI-TOF MS application to NTM isolates identification could positively impact on CML with a reduction in terms of cost per test from EUR 42.00 with LPA to EUR 1.50 [9,19,26]. The introduction of MBT Mycobacteria, a new commercially available ready-to-use kit, could enhance the per sample cost in comparison to the MycoEx protocol. However, even if an accurate cost-benefit analysis is needed to evaluate the MBT impact on NTM identification, the cost savings in comparison to LPA is largely preserved and the reproducibility performance could restrain the rising costs.

This work provides preliminary data about the recently introduced MBT Mycobacteria kit performance for both liquid and solid media, highlighting, for the first time, its strength and weakness in different conditions. Despite the promising MBT results, the study limitations include the number of analyzed NTM strains. Therefore, further studies with a higher number of NTM strains should be performed to confirm these preliminary results. Moreover, even if the analyzed NTM species are the principal aetiologic agents of mycobacteriosis, in-depth studies are needed to evaluate MBT in a wider group of mycobacteria species and subspecies (i.e., *Mycobacterium tuberculosis* complex, *M. abscessus subsp. bolletii*, *M. kansasii*, *M. chelonae*) and to assess its real-life TAT and cost-benefit impact.

## 5. Conclusions

To date, NTM identification has been limited to a few specialized laboratories. The capacity of MALDI-TOF MS to detect clinically interesting NTM in a rapid, reliable and cost-effective manner could provide a significant improvement for all CML.

The data collected in our study showed excellent identification performances for the new MBT Mycobacteria kit, especially when applied on NTM strains grown on solid media; however, it is important to perform an in-depth evaluation of a higher number of mycobacteria, in order to assess its effectiveness in mycobacteria identification and with CML workflows and workloads.

## Figures and Tables

**Figure 1 microorganisms-11-00120-f001:**
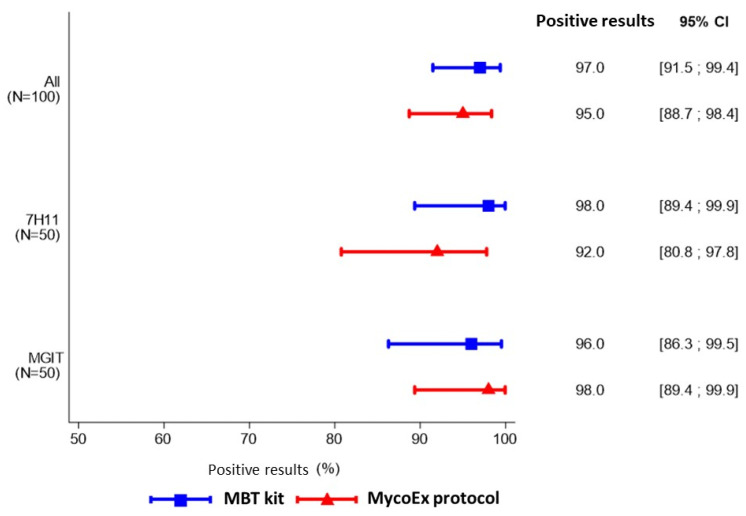
Availability of identification results of MBT Mycobacteria kit and MycoEx preparation protocol according to the culture media.

**Figure 2 microorganisms-11-00120-f002:**
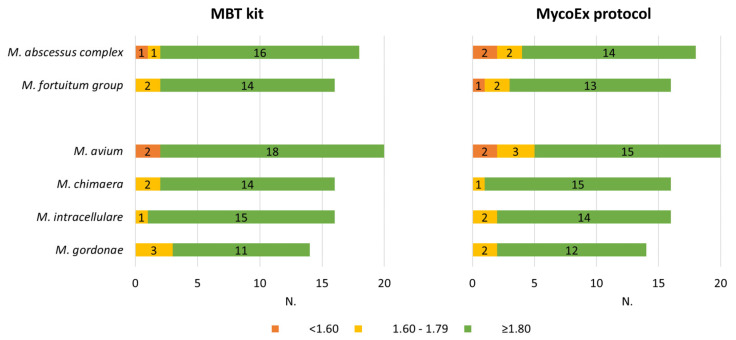
MALDI-TOF MS *log*(score) results in relation to the preparation treatment and the analyzed species or groups.

**Table 1 microorganisms-11-00120-t001:** Identification results probability, *log*(score), according to the extraction method, the culture media and the rapid and slow mycobacteria growth rate.

			7H11			MGIT	
		≥1.80	1.60–1.79	<1.60	≥1.80	1.60–1.79	<1.60
Extraction Protocol	N.	N. (%)	N. (%)	N. (%)	N. (%)	N. (%)	N. (%)
**MBT Mycobacteria kit**	50	47 (94.0)	2 (4.0)	1 (2.0)	41 (82.0)	7 (14.0)	2 (4.0)
RGM	17	15 (88.2)	1 (5.9)	1 (5.9)	15 (88.2)	2 (11.8)	0 (0.0)
SGM	33	32 (97.0)	1 (3.0)	0 (0.0)	26 (78.8)	5 (15.2)	2 (6.1)
**MycoEx Protocol**	50	42 (84.0)	4 (8.0)	4 (8.0)	41 (82.0)	8 (16.0)	1 (2.0)
RGM	17	12 (70.6)	2 (11.8)	3 (17.6)	15 (88.2)	2 (11.8)	0 (0.0)
SGM	33	30 (90.9)	2 (6.1)	1 (3.0)	26 (78.8)	6 (18.2)	1 (3.0)

7H11: Middlebrook 7H11 agar; MGIT: BBL™ MGIT broth; RGM: rapid growing mycobacteria; SGM: slow growing mycobacteria.

## Data Availability

Not applicable.

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
