# Peer review of "Evaluation of Two Different Preparation Protocols for MALDI-TOF MS Nontuberculous Mycobacteria Identification from Liquid and Solid Media"

_microorganisms, 2023, doi:10.3390/microorganisms11010120_

Round 1
Reviewer 1 Report
The study by Pastrone et al. evaluated two different preparation protocols for MALDI-TOF MS-based nontuberculous Mycobacteria identifications. Although the novelty of this study is not sound, the manuscript itself has been prepared in a good condition and the results and conclusion seem also clear enough to be considered for publication in Microorganisms. Following are some suggestions for authors to improve this manuscript.
1) Some more explanations as to why the optimizations of sample preparations are needed should be considered in the introduction.
2) it is strange to see many separated paragraphs in both Materials and Methods and Discussion sections. Reorganization and restructuring are needed. It might be better to use subsections to show clearly what the authors want to see in each section.
3) There might be some misunderstanding but how many replications are used in this study. P-value is required for each comparison.
Author Response
1) Some more explanations as to why the optimizations of sample preparations are needed should be considered in the introduction.
In lines 75-80 a description of MycoEx protocol’s main drawbacks and the potential advantages of MBT Mycobacteria kit was added.
2) it is strange to see many separated paragraphs in both Materials and Methods and Discussion sections. Reorganization and restructuring are needed. It might be better to use subsections to show clearly what the authors want to see in each section.
In Materials and Methods 3 new subheadings were used to highlight MBT and MycoEx procedure steps and MALDI-TOF MS spectra acquisition and analysis. In the Discussion section, unsuitable paragraphs were removed, identification performance were merged in a dedicated subsection, and another subheading was added for TAT and cost-saving considerations.
3) There might be some misunderstanding but how many replications are used in this study. P-value is required for each comparison.
Each identification test was performed in replicate for all extraction protocols and media. Considering the 50 positive samples, a total of 600 spectra were acquired by MALDI-TOF. p-values were added in the reproducibility evaluation (lines 181 and 184).
Reviewer 2 Report
Evaluation of two different preparation protocols for MALDI- 2 TOF MS Nontuberculous Mycobacteria identification from liquid and solid media by Pastrone et al., is a timely investigation as MALDI-TOF is being applied to identify MTB and Mold. Well written paper with few questions and concerns as below.
Major
Line 100- as you are trying to prove the effectivity of MBY for the first time, no literature supporting the safety of the inactivation protocol. Unless this reagent inactivation is approved by a governing body ( FDA or European counterpart (cite that), authors need a minor experiment to plate reagent inactivated samples to try and culture to show no growth.
Line 140- Table.1 add abbreviation to table legends to make life easy for the readership
Line 142- available is not a good word to describe the outcome. Is this percent agreement or present positive? Also in Fig 1, replace availably of identification results with percent positive or agreement percent.
Line 148- results in the 100% should be changed to 100% agreement
Fig.1. Bar colors codes are confusing and make different coding like checked or doted etc.
Line 157- summarize and provide false negative percentage from table A1 data.
Minor issues
Line 56- trough is a typo for through
Line 70 – replace argue with resolve
Author Response
Line 100- as you are trying to prove the effectivity of MBY for the first time, no literature supporting the safety of the inactivation protocol. Unless this reagent inactivation is approved by a governing body (FDA or European counterpart (cite that), authors need a minor experiment to plate reagent inactivated samples to try and culture to show no growth.
As declared, the study aim was to investigate the potential diagnostic performance of the application of MBT and MycoEx to the MALDI-TOF MS identification of the main clinically significant NTM. The extraction protocols efficacy in terms of mycobacteria cell inactivation wasn’t analysed. Preliminary inactivation results of MBT kit were presented in ECCMID 2022, poster n. P1033 “Performance of Easy MycoEX Method for MALDI Biotyper Identification of Mycobacterium Cultures from Liquid and Solid Media” M. Timke et al. Authors reported a 100% inactivation efficacy on TB isolates. However, in both protocols Bruker underlines that the efficacy of cell-inactivation is linked to the amount of tested biomass (“… it is strongly recommended to consider the samples as potentially infective until sample extract was overlaid with Bruker Matrix HCCA, portioned. Sample preparation should always be performed in a suitably secure environment according to local regulations.”). Moreover, in our work, we considered only NTM (BLS-2) and not on MTB (BLS-3).
Line 140- Table.1 add abbreviation to table legends to make life easy for the readership
In table 1 and table A1 abbreviations were reported in tables footnotes
Line 142- available is not a good word to describe the outcome. Is this percent agreement or present positive? Also in Fig 1, replace availably of identification results with percent positive or agreement percent.
Thank you for the observation, in line 142 and figure 1 is reported the number of positive results. The manuscript (both text and figure) was corrected (line 152, figure 1)
Line 148- results in the 100% should be changed to 100% agreement
The manuscript was corrected (lines 158-160)
Fig.1. Bar colors codes are confusing and make different coding like checked or doted etc.
Can you, please, confirm that problems are on figure 1? We suppose that they are related to figure 2. In fact in figure 1 colours (blue and red) and shapes (triangle and square) are both only linked to the extraction protocols. On the contrary, in figure 2, the colours, orange and yellow are too similar and near. We agree that the bar colours code could results confusing. For this reason, figure 2 was modified according to your suggestions.
Line 157- summarize and provide false negative percentage from table A1 data.
The requested information was added (lines 168-170)
Line 56- trough is a typo for through
Line 70 – replace argue with resolve
Corrections in line 56 and 70 were performed (new positions: lines 58 and 72)